# Ensiled Mixed Vegetables Enriched Carbohydrate Metabolism in Heterofermentative Lactic Acid Bacteria

**Daniel L. Forwood [1], Devin B. Holman [2], Sarah J. Meale [1,\*] and Alex V. Chaves [3,\*]**

[1] School of Agriculture and Food Sciences, Faculty of Science, The University of Queensland, Gatton, QLD 4343, Australia

[2] Lacombe Research and Development Centre, Agriculture and Agri-Food Canada, Lacombe, AB T4L 1V7, Canada

[3] School of Life and Environmental Sciences, Faculty of Science, The University of Sydney, Sydney, NSW 2006, Australia

\* Correspondence: s.meale@uq.edu.au (S.J.M.); alex.chaves@sydney.edu.au (A.V.C.); Tel.: +61-7-5460-1614 (S.J.M.); +61-2-9036-9312 (A.V.C.)

**Abstract:** This study evaluated the fermentation quality, nutritive profile, in vitro fermentation, and microbial communities colonising sorghum ensiled with an unsalable vegetable mixture (chopped beans, carrot, and onion (1:1:1) ) including: (1)−100% sorghum; (2)−80% sorghum + 20% vegetable mix or (3)−60% sorghum + 40% vegetable mix, on a dry matter (DM) basis, with or without a probiotic inoculant. Samples were obtained across 0, 1, 3, 5,7, and 101 days ensiling and after 14 d aerobic exposure. The V4 region of the 16S rRNA gene and the ITS1 region were sequenced to profile bacterial, archaeal, and fungal communities. Compared to the 0% DM, ethanol increased ($p < 0.01$) from 8.42 to $20.4 \pm 1.32$ mM with 40% DM vegetable mix inclusion, while lactate decreased from 5.93 to $2.24 \pm 0.26$ mM. Linear discriminant analysis revealed that relative abundances of 12 bacterial taxa were influenced by silage treatments (log LDA score $\geq 4.02$; $p \leq 0.03$), while predicted functional pathways of alternative carbohydrate metabolism (hexitol, sulfoquinovose and glycerol degradation; N-acetyl glucosamine biosynthesis; log LDA score $\geq 2.04$; $p \leq 0.02$) were similarly enriched. This study indicated that carbohydrate metabolism by heterofermentative lactic acid bacteria can increase the feed value of sorghum when ensiled with an unsalable vegetable mixture at 40%DM, without requiring a high quantity of lactate.

**Keywords:** 16S rRNA sequencing; unsalable vegetable silage; lactic acid bacteria; silage microbial profile; aerobic stability

## 1. Introduction

In Australia, between 40–50% of food waste occurs at the consumer level, with fruit and vegetables wasted at a greater rate than other food categories [1]. Compounding this issue, an additional 3300 kilotonnes are generated at the pre-consumer processing stage in the Australian horticultural industry [2], with vegetables comprising the greatest volume of produce disposed of into landfills [3]. To reduce this magnitude of wastage other methods of utilisation, such as anaerobic digestion for energy production, have been used but are not widely adopted in Australia [4].

The anaerobic fermentation of plant material via ensiling for ruminant feed production presents a twofold opportunity to utilise vegetable waste while value-adding low protein crops such as maize or sorghum which range from 10.6–11.6% dry matter (DM), respectively compared to lucerne at 15.3% DM [5]. Previously, ensiling maize with up to 40% DM carrot or pumpkin decreased neutral detergent fibre (NDF) by 6.72%, increased crude protein by 34.9%, and in vitro dry matter digestibility (IVDMD) by 6.63%, indicating promise as a ruminant feed [6], with similar effects also noted in sorghum [7].

However, little is known regarding the feed value or silage quality of other unsalable vegetables, as studies have predominantly focused on vegetable waste. For example, green bean waste fermented for 33 days and inoculated with *Lactiplantibacillus plantarum* had greater acetate and lactate concentrations than potato waste silages (216 vs. 118; 388 vs. 306 µM/g dry weight) [8]. However, the use of a vegetable mixture comprising carrot, cabbage, beet, and onion, promoted fermentation conditions conducive to the growth of *Lactobacillus* and *Leuconostoc* spp., producing high quantities of lactate, and lactate and acetate, respectively [9]. These microbes and acids are integral to silage fermentation quality [10]. Similarly, gradually increasing the green bean content within maize silage increased crude protein content similar to that of clover or lucerne silages in chemical composition [11], while decreasing NDF and methane ($CH_4$) yield [12].

The aim of this study was to evaluate the influence of ensiling sorghum with 20 or 40% DM unsalable vegetable mix [chopped beans, carrot, and onion (1:1:1)], with or without a soil probiotic inoculant on silage and in vitro fermentation parameters, archaeal, bacterial, and fungal microbial communities across the ensiling period, as well as silage aerobic stability. It was hypothesized that ensiling a mixture of vegetables with sorghum would enhance silage fermentation resulting in lactic and acetate production from dominant *Lactobacillus* spp. while producing silage of similar chemical composition to conventional legume silages.

## 2. Materials and Methods

### 2.1. Silage Production and Sampling

Silage was produced per the methods of [7]. Briefly, crop sorghum was harvested from The University of Queensland (Gatton QLD) in January 2020. Beans, carrots, and onions deemed unsalable for retail sale were sourced from Kalfresh (Kalbar QLD) and segmented into halves of approximately $5 \times 5 \times 2$ cm in size in preparation for ensiling. Chopped beans, carrots, and onions (1:1:1) were mixed with crop sorghum. Silage treatments comprised (1) 100% crop sorghum: (2) 20% vegetable mixture + 80% crop sorghum and (3) 40% vegetable mixture + 60% crop sorghum on a DM basis. Silage treatments were packed into thirty individual PVC mini-silos to a density of 240 kg/m$^3$, resulting in two mini-silos per treatment.

Similarly, thirty PVC mini-silos were packed with the same silage material but were treated with 5 mL Bio Soil probiotics administered in an aerosolised state (Bio Soil®, OzProbiotics, Penrith, NSW, Australia). The Bio Soil contained: *Bacillus subtilis*, *Bifidobacterium animalis*, *Bifidobacterium bifidum*, *Bifidobacterium longum*, *Enterococcus lactis*, *Streptococcus thermophilus*, *Lactobacillus acidophilus*, *Lactobacillus delbrueckii subsp. bulgaricus*, *Lactobacillus casei*, *Limosilactobacillus fermentum*, *L. plantarum*, and *Saccharomyces cerevisiae* prior to ensiling. These treatments are considered probiotics.

Silage was sampled at 0, 1, 3, 5, 7, and 101 d ensiling. Upon opening, 500 g silage was randomly selected, placed on an aluminium tray, and oven-dried at 65 °C until a constant weight for DM analysis and chemical composition. Random samples were collected from mini silos, where approximately 200 g designated for DNA extraction was wrapped in aluminium foil and stored at −20 °C for downstream extraction and sequencing.

### 2.2. Chemical Composition

The silage was collected after 101 d ensiling and analysed for chemical composition. Samples designated for determination of ash content were dried for 24 h at 100 °C to determine analytical DM, then placed in a furnace at 500 °C for 6 h [AOAC (942.05 2002)]. Silage samples were ground and sieved through a 1 mm screen for neutral detergent fibre (NDF) content analysis using the methods of [13], modified for the use of an ANKOM 200/220 Fiber Analyzer (ANKOM Technol. Corp., Fairport, NY, USA). Sodium sulphite and amylase were incorporated into NDF analyses and included residual ash (aNDFom). Crude fat of feed was determined by extraction of ether for lipid extraction [AOAC (929.29 1995)] with modifications for an ANKOM Fat Analyzer (ANKOM Technol. Corp., Fairport, NY,

USA). Silage nitrogen content was determined using a LECO CN928 Carbon/Nitrogen combustion analyser (Leco, St Joseph, MI, USA). The instrument was set up per manufacturer recommendations, where the instrument was standardised with EDTA and combustion temperature set to 1100 °C. Approximately 0.3 g ground silage sample was weighed into a ceramic boat and analysed. Crude protein (CP) was calculated as mineral nitrogen × 6.25. Non-fibre carbohydrates (NFC) were calculated per the methods of [14] where:

$$NFC = 100 - CP - NDF - \text{crude fat} - \text{ash} \qquad (1)$$

### 2.3. pH, Volatile Fatty Acids, and Organic Acids

Approximately 15 g of silage sample was collected at each sampling period, combined with 135 mL distilled water, and blended at room temperature for 30 s. After filtering through cheesecloth, the pH was measured with 15 mL of filtrate using a Hanna Edge HI2002 pH meter (Hanna Instruments, Woonsocket, RI, USA. The remaining filtrate (30–40 mL) was placed on ice, then centrifuged at 10,000× *g* for 15 min at 4 °C. Subsequent 5 mL replicates were combined with 1 mL metaphosphoric acid (5:1 ratio *v/v*) and stored at −20 °C until analysis for volatile fatty acids (VFA) and organic acids.

Volatile fatty acids were analysed per the methods of [15]. Briefly, a 1.5 mL sample was centrifuged at 12,000 rpm for 2 min. An aliquot of 1.2 mL was combined with 0.2 mL crotonic acid solution, stood at room temperature, and centrifuged for a further 10 min at 12,000 rpm. The supernatant was removed to an autosampler vial for analysis of VFA and organic acids via gas chromatography per the methods of [7]. Briefly, malonic acid (50 mM) was used as an internal standard for organic acids and ethanol analyses. The concentrations of organic acids and VFA were expressed in mM and ethanol was expressed as a percentage.

### 2.4. Profiling of the 16S rRNA and ITS1 Microbial Communities

DNA from samples collected on days 0, 1, 3, 5, 7, 101 ensiling, and at the end of the aerobic stability period were extracted via the bead-beating method of [16] with extraction and on-column purification using the Qiagen DNEasy Powerlyzer Powersoil kit (Qiagen, Hilden, Germany), and DNA concentration quantified using a Nanodrop 1000 Spectrophotometer (Thermo Fisher Scientific, Waltham, MA, USA). Approximately 15 µg of DNA was sent to Génome Québec (Montréal, QC, Canada) for fluidigm amplification and MiSeq Illumina sequencing (Illumina Inc., San Diego, CA, USA). The 515F (5′-GTGYCAGCMGCCGCGG TAA-3′) and 806R (5′-GGACTACNVGGGTWTC TAAT-3′) primers were selected for amplification targeting the V4 region of the 16S rRNA gene of archaea and bacteria per the methods of [7], while the ITS1F (5′-CTTGGTCATTTAGAGGAAG TAA-3′) and ITS2 (5′-GCTGCGTTCTTCATCGATGC-3′) primers were selected to target the fungal ITS1 region.

The DADA2 v. 1.8 [17] package was used in R v. 4.0.2 [18] within RStudio 1.3.959 [19] to process the 16S rRNA gene and ITS1 region sequences. Briefly, the forward and reverse 16S rRNA gene sequences were trimmed to 220 and 200 bp, respectively, and primers and chimeras were removed. Short sequences were subsequently filtered out if ≤50 bp. The sequences were merged, amplicon sequence variants (ASVs) were resolved, and taxonomy was assigned to these ASVs using the SILVA v. 138.1 database [20]. ITS1 sequences were not trimmed, but a minimum sequence length of 50 bp was used. The ITS1 sequences were merged, chimeras were removed, and taxonomy was assigned to these ITS1 ASVs using the UNITE v. 10.05.2021 database [21]. Alpha diversity metrics (Chao1, Shannon, and inverse Simpson) for the 16S rRNA gene and ITS1 datasets were calculated in R using Phyloseq v. 3.14 [22]. Beta diversity was assessed with Bray–Curtis dissimilarities calculated using vegan 2.5–7 [23] in R, and the effects of the ensiling period, vegetable inclusion level, and probiotic inoculation were calculated through PERMANOVA.

The predicted functional composition of the bacterial communities in vegetable mix silages was determined using the bacterial ASV sequences in PICRUSt2 v. 2.4.2 ([24];

(https://github.com/picrust/picrust2, accessed on 19 August 2021)), and the MetaCyc database (https://metacyc.org/, accessed on 19 August 2021). For the purpose of this study, MetaCyc pathways are closer to true biological pathways than KEGG maps given they do not comprise reactions from multiple organisms in nature [25].

All 16S rRNA gene and ITS1 region sequences were submitted to the Sequence Read Archive under BioProject PRJNA821258.

### 2.5. Aerobic Stability

After opening, a sub-sample of approximately 500 g of silage was placed into an aluminium tray and exposed to laboratory conditions at a constant temperature for 14 d. On consecutive days, measurements of silage and mean room temperature ($^\circ$C) were collected twice daily using a FLIR E50 thermal imaging camera (FLIR, Wilsonville, USA), and FLIR Tools software was used. Aerobic stability was defined [26] as the number of hours of aerobic exposure prior to an increase of silage temperature greater than 2 $^\circ$C above ambient temperature. After the aerobic stability period, samples (one per mini silo) were mixed, and a subsample of approximately 70 g was collected and freeze-dried for DNA extraction.

### 2.6. In Vitro Rumen Fermentation, Gas, and Methane Production

Rumen contents designated for in vitro batch culture fermentation, gas, and $CH_4$ production parameters were collected from rumen-cannulated Holstein steers ($n = 3$) housed at The University of Queensland Gatton Dairy under the approval of The University of Queensland Animal Ethics Committee (Approved Protocol Number 2021/AE000823). Rumen liquor was collected per the methods of [27], and preparation of inoculum for batch culture was prepared according to [28].

Incubation, gas and $CH_4$ production sampling occurred via the methods of [7] with adjustments. Briefly, bottles were warmed for an hour prior to incubation in a Ratek OM25 Digital Shaking Incubator (Ratek Instruments Pty Ltd., Boronia, Australia), set to 39 $^\circ$C. Bottles were subsequently filled with 25 mL inoculum under a stream of $CO_2$, sealed, and returned to the incubator set to 90 rpm for 24 h. Each incubation was repeated in triplicate, including blank samples containing 25 mL inoculum only.

Gas production was measured using a water displacement apparatus [29], while pH was determined via the methods of [27]. At the end of the incubation period, bags were removed from bottles and immediately placed on ice to cease fermentation, rinsed with distilled water, and oven-dried at 65 $^\circ$C until the weight was consistent. Dried bags were weighed for the calculation of IVDMD.

### 2.7. Statistical Analysis

Silage chemical composition, fermentation parameters, VFA, organic acids, and measures of alpha diversity were analysed as a completely randomized design using the MIXED procedure of SAS ([30]; SAS Online Doc 9.1.4) with fixed effects vegetable level (0, 20 and 40%), with or without probiotics, and their interactions. Mini silo within treatment was considered a random effect. Further, the relative abundance ($\geq$0.1%) of bacterial and fungal genera relative abundance detected in >15% of samples were considered biologically relevant and analyzed by the MIXED procedure of SAS using the same fixed and random effect parameters.

In vitro fermentation, VFA, gas, and $CH_4$ production were analyzed as described above, but random effects were defined as run and run $\times$ treatment. Similarly, aerobic stability was analysed via the MIXED method, instead using the random effects of day and day $\times$ treatment (temperature). All results were expressed as LSMEANS with standard error of the mean (SEM), with significance declared when $p \leq 0.05$, and tendencies reported when $0.05 < p \leq 0.10$. Experimental units in this study consisted of individual mini silos and incubation run for the in vitro fermentation, respectively.

Testing for normal distribution of the data was conducted using the UNIVARIATE procedure of SAS, while the effect of vegetable level and probiotic application of the silage microbial community structure was determined using PERMANOVA (adonis2 function) and Bray-Curtis dissimilarities in R using the vegan 2.5–6 package [23]. Further, the Galaxy platform (v.1.39.5.0; https://galaxyproject.org/, accessed on 19 August 2021; [31]) was used to calculate the linear discriminant analysis effect size (LEfSe) (https://huttenhower. sph.harvard.edu/galaxy/, accessed on 19 August 2021; [32]) to ascertain the bacterial and fungal taxa likely to explain treatment effects noted across analysed silage parameters. Moreover, if LDA scores were $\geq 2.0$ and $p \leq 0.05$, they were considered significant, and thus, differentially expressed [33]. LEfSe was also used to identify MetaCyc metabolic pathways that were predicted to be more enriched in certain vegetable mixtures.

Spearman correlation coefficients were calculated between silage fermentation parameters, differentially expressed bacterial taxa and significant functional pathways were subsequently visualised using corrplot on R.

## 3. Results

### 3.1. Chemical Composition

An interaction between vegetable level and probiotic was observed for crude protein which was greater ($p = 0.05$) in inoculated 0% DM vegetable mix silages, compared to other silage samples (Table 1). However, there was no effect of level × probiotic on other silage treatments, although inoculation tended ($p = 0.08$) to reduce DM content by 6%, with 0% DM silages, compare to those containing vegetable mix. Further, no effect of probiotic inoculation was observed on silage composition, except ash, which tended ($p = 0.08$) to be 9% higher with inoculation (12.1 vs. 13.2 $\pm$ 0.37% DM; $p = 0.08$).

**Table 1.** Dry matter (DM) content and chemical composition of sorghum-vegetable mixture silages containing 0, 20, or 40% vegetable mixture on a DM basis.

| Probiotic (Prob) | NO | | | YES | | | | *p*-Value | | | | |
| --- | --- | --- | --- | --- | --- | --- | --- | --- | --- | --- | --- | --- |
| Level, % DM | 0 | 20 | 40 | 0 | 20 | 40 | SEM | Level | Prob | Level × Prob | L | Q |
| DM content, % | 38.2 | 25.6 | 17.4 | 35.9 | 26.7 | 18.4 | 0.69 | <0.01 | 0.95 | 0.08 | <0.01 | 0.07 |
| CP, % DM | 9.16 [a] | 12.3 [a] | 8.47 [a] | 14.8 [a] | 10.7 [b] | 8.63 [b] | 1.17 | 0.05 | 0.20 | 0.05 | 0.03 | 0.26 |
| NFC, % DM | 20.7 | 19.8 | 26.4 | 12.8 | 21.4 | 23.8 | 2.79 | 0.06 | 0.24 | 0.30 | 0.02 | 0.91 |
| EE, % DM | 7.07 | 5.62 | 5.99 | 6.78 | 6.51 | 6.68 | 0.35 | 0.11 | 0.18 | 0.27 | 0.14 | 0.11 |
| NDF, % DM | 51.2 | 49.6 | 47.3 | 52.8 | 48.6 | 46.9 | 2.43 | 0.21 | 0.98 | 0.86 | 0.09 | 0.84 |
| Ash, % DM | 11.9 | 12.6 | 11.8 | 12.9 | 12.8 | 13.9 | 0.64 | 0.77 | 0.08 | 0.38 | 0.49 | 0.95 |

Abbreviations: CP, crude protein; NFC, non-fibre carbohydrates; EE, ether extract; NDF, neutral detergent fibre; SEM, standard error of the mean; L, linear; Q, quadratic. Different letters (a, b) within the rows indicate differences ($p < 0.05$).

### 3.2. Silage Organic Acids and Volatile Fatty Acids

#### 3.2.1. Silage after 101 d Ensiling

Silage pH increased by 12% with 20% DM vegetable inclusion ($p = 0.01$; Table 2) and was similarly increased with inoculation (level × probiotic; Table S1). Ethanol tended ($p = 0.06$; Table S1) to be influenced by vegetable level × probiotic and increased from 6.86 mM in 0% to 22.9 $\pm$ 1.87 mM in inoculated 40% DM vegetable mix silages. An effect of the vegetable level was also evident, where ethanol concentration linearly increased ($p < 0.01$; Table 2) up to 20.4 $\pm$ 1.32 with 40% DM vegetable mix inclusion. There was no effect of vegetable mix inclusion, except for lactate, which linearly declined by up to 62% with 40% DM vegetable inclusion ($p < 0.01$; Table 2). Similarly, no influence ($p \geq 0.11$) of level, probiotic, or level × probiotic was observed for VFA except for the minor VFA, which was reduced by 77% (9.38 $\pm$ 1.97 vs. 2.18 $\pm$ 1.85; $p = 0.03$; data not presented) with probiotic inoculation, compared to uninoculated silages. The percentage of propionate of total VFA tended ($p = 0.07$) to increase with 20% DM vegetable inclusion and was reduced by 17% with inoculation compared to the uninoculated 0% DM treatment (Table 2).

**Table 2.** Silage pH, percentages, and concentrations of volatile fatty acids (VFA) in sorghum-vegetable mixture silages at day 101 containing 0, 20, or 40% vegetable mixture on a dry matter basis.

| | Level, % DM | | | | *p*-Value | | |
|---|---|---|---|---|---|---|---|
| | **0** | **20** | **40** | **SEM** | **Level** | **L** | **Q** |
| **pH** | 3.9 [c] | 4.3 [b] | 4.4 [a] | 0.17 | <0.01 | <0.01 | <0.01 |
| **Organic acids, mM** | | | | | | | |
| Lactate | 5.93 [a] | 3.43 [b] | 2.24 [c] | 0.23 | <0.01 | <0.01 | 0.07 |
| Succinate | 0.39 | 0.42 | 0.40 | 0.02 | 0.39 | 0.70 | 0.19 |
| Ethanol | 8.42 [b] | 12.7 [b] | 20.4 [a] | 1.32 | <0.01 | <0.01 | 0.34 |
| **Volatile fatty acids** | | | | | | | |
| Total VFA, mM | 15.2 | 17.2 | 18.0 | 1.78 | 0.54 | 0.29 | 0.79 |
| *Volatile fatty acids as a percentage of total VFA* | | | | | | | |
| Acetate, % | 77.6 | 85.7 | 87.5 | 3.47 | 0.17 | 0.08 | 0.50 |
| Propionate, % | 3.61 | 5.07 | 4.72 | 0.37 | 0.07 | 0.07 | 0.09 |
| Butyrate, % | 8.41 | 4.60 | 5.43 | 1.27 | 0.16 | 0.14 | 0.19 |
| Minor VFA, % | 10.3 | 4.67 | 2.32 | 2.34 | 0.11 | 0.04 | 0.59 |

Abbreviations: Minor VFA, branch chained volatile fatty acids (iso-butyrate + iso-valerate), caproate and valerate; SEM, standard error of the mean. Different letters (a, b, c) within the rows indicate differences ($p \leq 0.05$). L, linear; Q, quadratic. There was no effect of level × probiotic ($p \geq 0.16$) for all parameters, but there was a single effect of probiotic on minor VFA ($p = 0.03$).

### 3.2.2. Fermentation Acids during the Ensiling Period

When separated by ensiling day, there was no effect ($p \geq 0.16$) of vegetable level, probiotic, or vegetable level × probiotic on the concentrations of acetate or total VFA, except on day 0, where inoculation with the probiotic increased ($p = 0.03$; Table S1) acetate from an undetected level to $0.21 \pm 0.05$ compared to uninoculated pre-ensiled samples, and vegetable mix inclusion increased ($p = 0.01$) total VFA from 1.56 to $2.22 \pm 0.12$ mM, respectively.

Ethanol concentration increased ($p \leq 0.02$) with 40% DM vegetable mix compared to the 0% DM silages on days 1 (2.74 vs. $5.08 \pm 0.57$ mM), 5 (3.98 vs. $11.2 \pm 0.61$ mM; Table S1) and 101 (8.42 vs. $20.4 \pm 1.32$ mM), while inoculation resulted in an increase ($p = 0.02$) from 2.29 to $4.32 \pm 0.47$ mM compared to uninoculated samples or pre-ensiled samples. An increase was also noted for succinate and lactate on day 5, increasing ($p < 0.01$) from 0.34 to $0.48 \pm 0.02$ mM and 4.88 to $8.31 \pm 0.47$ mM, respectively, but lactate also decreased ($p < 0.01$) from 5.93 to $2.24 \pm 0.23$ mM after 101 d ensiling. Moreover, succinate tended ($p = 0.06$) to be greater in inoculated, rather than uninoculated pre-ensiled samples.

Ethanol, lactate, and succinate concentrations were impacted by vegetable level × probiotic, where ethanol was greatest ($23.3 \pm 1.22$ mM; $p = 0.01$; Table S1) in inoculated 40% DM mix compared to all other treatments after 7 d ensiling, while a tendency ($22.9 \pm 1.87$ mM; $p = 0.06$) of this effect was similarly observed at d 101. Lactate and succinate concentrations were greatest after 1 d ensiling in uninoculated 0% DM ($3.59 \pm 0.27$ mM; $p = 0.05$) and inoculated 20% DM vegetable mix treatments ($0.49 \pm 0.02$ mM; $p = 0.01$), respectively. Moreover, inoculation inclusion of 40% DM vegetable mix increased ($p < 0.01$) d 101 silage pH to 4.4 vs. $3.9 \pm 0.02$ for 0% DM samples, irrespective of inoculation.

### 3.3. Bacterial, Archaeal, and Fungal Alpha Diversity Indices

During the ensiling period, the Chao1 richness estimator was only affected by level × probiotic after 5 d ensiling, where values for uninoculated silages were greatest with 40% DM vegetable mix ($56 \pm 2.41$), a similar effect was observed in inoculated silages with 20% DM vegetable mix ($45.5 \pm 2.41$; $p = 0.02$; Table S2). However, the level increased ($p \leq 0.05$) Chao1 after 1, 3, and 7 d ensiling with greater proportions of mixed vegetables. Pre-ensiled material had a greater inverse Simpson diversity with 20% DM vegetable mixture, but after 3 and 5 d ensiling, this was increased up to 18.4% with inoculated silages at 40% DM vegetable mix inclusion (level × probiotic; $p \leq 0.04$).

Likewise, pre-ensiled material with 20% DM vegetable mix had a 25% higher Shannon diversity index value than the 0% DM material, while values were greatest with 40% DM mix after 5 d ensiling ($p \leq 0.01$; Table S2). However, after 3 d ensiling, inoculation and 40% DM vegetable mix had a 16.9% higher Shannon diversity index value ($p \leq 0.01$). No effect of probiotic, or level × probiotic ($p \geq 0.11$; Table S2) was observed for bacterial or archaeal measures of alpha diversity across pre-ensiled, ensiled, or communities after aerobic exposure for 14 d. In contrast, inverse Simpson's and Shannon diversity indices were greatest with 20% vegetable mix inclusion compared to 0%, increasing respectively in pre-ensiled communities ($p = 0.01$; Table 3). No other effect ($p \geq 0.13$) of the level was observed across pre-ensiling, ensiling, or aerobic exposure periods.

**Table 3.** Measures of alpha diversity for the bacterial, archaeal, and fungal communities of vegetable mix silages ensiled with 0, 20, or 40% DM vegetable mix pre-ensiled, 101 d ensiled, and after 14 d aerobic exposure.

| | Level, % DM | | | | *p*-Value | | |
|---|---|---|---|---|---|---|---|
| | **0** | **20** | **40** | **SEM** | **Level** | **L** | **Q** |
| **Bacteria and archaea** | | | | | | | |
| *Pre-ensiled* | | | | | | | |
| Chao1 | 47.0 | 49.8 | 40.3 | 2.86 | 0.13 | 0.15 | 0.13 |
| Inverse Simpson | 3.61 [b] | 5.74 [a] | 4.10 [b] | 0.34 | 0.01 | 0.34 | <0.01 |
| Shannon | 1.82 [b] | 2.28 [a] | 2.02 [b] | 0.06 | 0.01 | 0.06 | <0.01 |
| *101 d Ensiled* | | | | | | | |
| Chao1 | 102.8 | 88.3 | 69.0 | 12.7 | 0.25 | 0.11 | 0.88 |
| Inverse Simpson | 6.01 | 4.99 | 5.01 | 0.63 | 0.47 | 0.30 | 0.53 |
| Shannon | 2.41 | 2.22 | 2.26 | 0.15 | 0.69 | 0.53 | 0.58 |
| *14 d aerobic exposure* | | | | | | | |
| Chao1 | 113 | 96.8 | 80.8 | 23.9 | 0.66 | 0.38 | 1.00 |
| Inverse Simpson | 10.3 | 8.29 | 7.93 | 4.37 | 0.92 | 0.71 | 0.88 |
| Shannon | 1.82 | 2.28 | 2.02 | 0.06 | 0.91 | 0.80 | 0.75 |
| **Fungi** | | | | | | | |
| *Pre-ensiled* | | | | | | | |
| Chao1 | 38.8 | 35.8 | 41.8 | 1.44 | 0.07 | 0.19 | 0.04 |
| Inverse Simpson | 4.26 | 3.95 | 4.63 | 0.27 | 0.28 | 0.37 | 0.18 |
| Shannon | 2.05 | 1.91 | 2.11 | 0.07 | 0.18 | 0.59 | 0.08 |
| *101 d Ensiled* | | | | | | | |
| Chao1 | 17.5 [a] | 8.75 [b] | 9.50 [b] | 1.59 | 0.01 | 0.01 | 0.05 |
| Inverse Simpson | 4.01 [a] | 1.47 [b] | 2.13 [b] | 0.52 | 0.03 | 0.04 | 0.05 |
| Shannon | 1.59 [a] | 0.52 [b] | 0.93 [ab] | 0.21 | 0.03 | 0.07 | 0.03 |
| *14 d aerobic exposure* | | | | | | | |
| Chao1 | 7.50 | 8.50 | 6.00 | 1.47 | 0.52 | 0.50 | 0.37 |
| Inverse Simpson | 2.66 | 2.02 | 2.19 | 0.33 | 0.43 | 0.35 | 0.36 |
| Shannon | 1.02 | 0.95 | 0.91 | 0.14 | 0.83 | 0.56 | 0.94 |

Abbreviations: SEM, standard error of the mean. No effect of probiotic ($p \geq 0.15$) was observed, except tendencies with Chao 1 and Inverse Simpson diversity index ($0.06 \geq p \geq 0.09$) after 14 d aerobic exposure. Similarly, effects were noted for level × probiotic for fungal communities, but no effect ($p \geq 0.01$) was observed for bacterial communities, thus this data was omitted. Letters (a, b) with the rows indicate differences ($p \leq 0.05$). L, linear; Q, quadratic.

Bacterial and archaeal measures of alpha diversity of pre-ensiled and silage after aerobic exposure for 14 d were influenced by the effect of level × probiotic ($p \leq 0.01$; Table 3). This was noted for the inverse Simpson diversity index, where pre-ensiled values were greater with 40% DM mix without the inoculant (5.76 vs. 3.77 ± 0.38; data not presented) and reduced by 40% DM mix inclusion and by probiotic inoculation (3.51 vs. 4.76 ± 0.38; $p = 0.01$; data not presented) against the control. A similar effect was also observed after 14 d aerobic exposure (2.47 vs. 1.28 ± 0.47 and 4.03 vs. 1.91 ± 0.47; $p = 0.02$; data not presented).

Fungal alpha diversity was unaffected by level, probiotic, or level × probiotic across the ensiling period, except on day 101, where Chao1 decreased by 51.5% with increasing vegetable level ($p = 0.01$; Table S3). However, the inverse Simpson and Shannon diversity indices were only affected by level × probiotic on d 0 and 101 of ensiling, where inoculation reduced both metrics when vegetable level increased, while the opposite was observed in

uninoculated samples ($p \leq 0.02$). Similarly, 20% DM vegetable mix decreased the Shannon diversity index by 67.3% ($p = 0.03$).

Uninoculated 40% DM mix samples had greater fungal Shannon diversity than the control (2.28 vs. 1.92 ± 0.10; Table 3) but inoculated 40% DM mix had lower Shannon diversity (1.93 vs. 2.18 ± 0.10; $p = 0.02$). There was a tendency of level regarding pre-ensiled Chao1 indices, where Chao1 increased with up to 40% DM mix by 7.7% ($p = 0.07$). The effect of the level was observed for all three alpha diversity metrics among fungal communities from samples ensiled for 101 d, decreasing with increasing vegetable mix inclusion ($p \leq 0.03$).

### 3.4. Bacterial and Archaeal Community Structure and Profile Abundance

Based on the Bray-Curtis dissimilarities and PERMANOVA, the structure of the silage microbiota was influenced by level × probiotic at 101 days ensiling (PERMANOVA: $R^2 = 0.18$; $p = 0.01$; Figure 1A). However, after 14 d aerobic exposure, no effect of level × prob was observed on the community structure of the silage microbiota (PERMANOVA: $R^2 = 0.14$; $p = 0.27$; Figure 1B).

Among the 59 archaeal and bacterial genera identified, *Weissella* had the highest relative abundance in uninoculated samples with 20% DM mix inclusion (52.7 ± 2.22%; $p < 0.01$; Table S4), while similarly dominating 0% mix samples with or without probiotic inoculant (36.9 vs. 39.9 ± 2.22%; $p < 0.01$). *Pediococcus* and *Lactiplantibacillus* dominated ($p < 0.01$) the bacterial and archaeal communities of 0% DM mix samples at 39.4 ± 0.84%, and 16.5 ± 0.82% compared to 20 or 40% DM mix, irrespective of probiotic, or the interaction of level × probiotic. However, *Lentilactobacillus* and *Lactobacillus* (41.8 ± 3.99% and 30.0 ± 2.67%; $p < 0.01$) were the most dominant taxa among inoculated 40% DM mix samples, while *Lactobacillus* dominated (71.7 ± 2.67%; $p < 0.01$) the uninoculated 40% DM mix silage microbiota. Tendencies (Table S4) were noted for *Serratia*, *Stenotrophomonas*, and *Chryseobacterium*, which decreased in relative abundance with inoculation, while *Serratia* was also reduced ($p = 0.07$) with increasing vegetable mix inclusion.

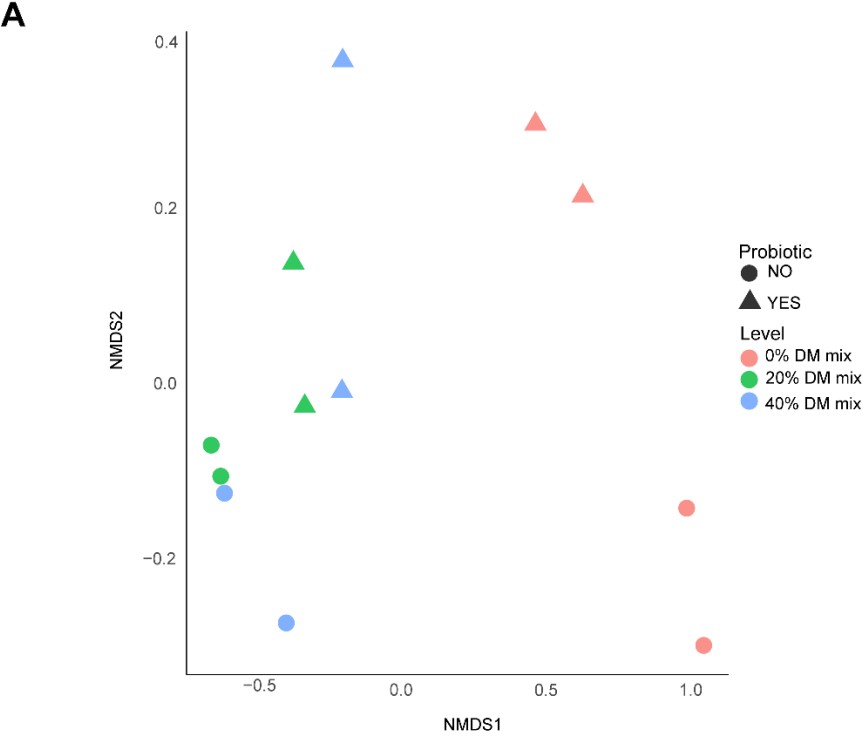

**Figure 1.** *Cont.*

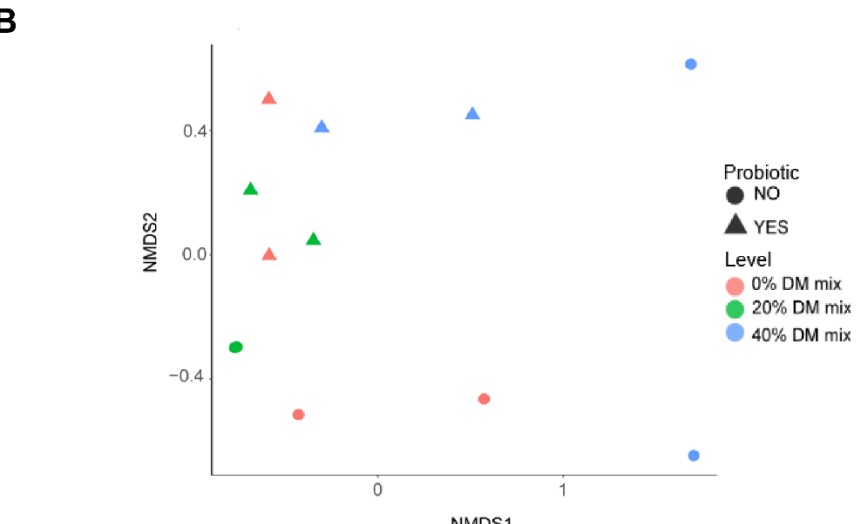

**Figure 1.** Non-metric multidimensional scaling (NMDS) plot based on the Bray–Curtis dissimilarities of the bacterial and archaeal microbiota of sorghum mixed with 0, 20 or 40% DM vegetable mix after (**A**) 101 d ensiling and (**B**) after 14 d aerobic exposure, with or without a probiotic inoculant.

The linear discriminant analysis effect size (LEfSe) analysis of silage samples indicated that the relative abundances of *Levilactobacillus, Leuconostoc, Lactococcus, Lactiplantibacillus,* and *Pediococcus* were significantly associated with the 0% DM sorghum silage microbiota (log LDA score ≥ 4.05; $p ≤ 0.03$; Figure 2A). However, 20% of DM vegetable mixtures were significantly associated with *Weissella, Sphingomonas, Bacillus,* and *Limosilactobacillus* (log LDA score ≥ 4.02; $p = 0.02$), while 40% of the DM vegetable mix was associated with *Achromobacter, Lactobacillaceae* HT002 and *Lactobacillus* (log LDA score = 4.41; $p ≤ 0.01$).

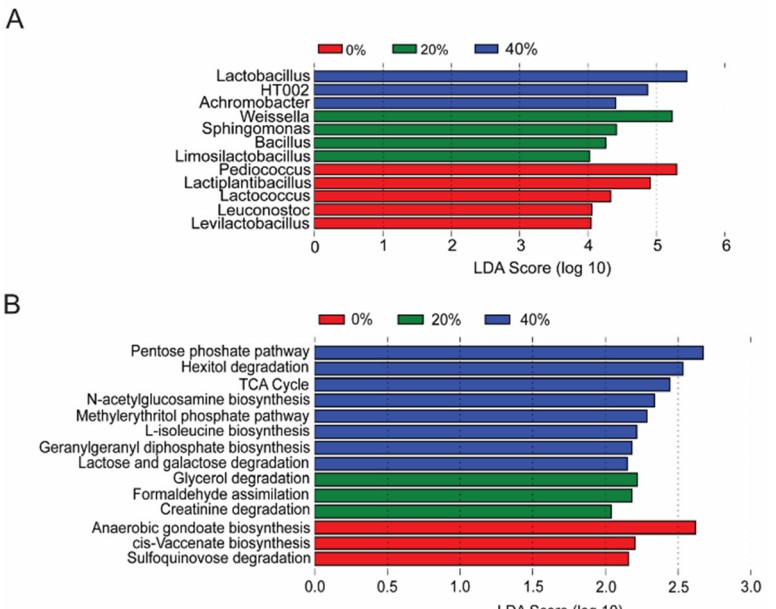

**Figure 2.** Linear discriminant analysis effect size (LEfSe) of (**A**) bacterial communities and (**B**) MetaCyc pathways for sorghum ensiled with 0, 20, or 40% DM vegetable mix. The effect of treatment ($p ≤ 0.05$) was identified if LDA log 10 score ≥ 2.0.

### 3.5. Differential Expression of Functional Pathways

The metabolic pathways sulfoquinovose degradation, anaerobic gondoate biosynthesis, and cis-vaccenate biosynthesis were predicted by PICRUSt2 and LEfSe to be enriched in

the 0% DM mix silage microbiome (log LDA score $\geq$ 2.16; $p \leq 0.02$; Figure 2B). However, the 20% DM mix had a predicted metagenome that was enriched with creatinine degradation, glycerol degradation, and formaldehyde assimilation pathways (log LSA score $\geq$ 2.04; $p \leq 0.02$), while the metagenomes of the ensiled 40% DM mix with sorghum was predicted to have a greater relative abundance of the pentose phosphate pathway, lactose and galactose degradation, N-acetylglucosamine biosynthesis, L-leucine biosynthesis, hexitol degradation, the citric acid cycle, methylerythritol phosphate pathway, and geranylgeranyl diphosphate biosynthesis genes (log LDA score $\geq$ 2.81; $p \leq 0.02$).

*3.6. Correlation Coefficients of Bacterial Communities, Fermentation Parameters, and Predicted Functional Pathways*

There were 145 significant correlations (120 positive; 25 negative) detected between bacterial communities, silage fermentation parameters, and predicted metabolic pathways. Of the significant correlations, ethanol, lactate, and silage pH were strongly correlated with *Lactobacillaceae* HT002, *Lactobacillus,* and *Levilactobacillus* ($-0.93 \leq R^2 \leq 0.88$; $p \leq 0.04$; Figure 3), while these parameters were similarly correlated with the sulfoquinovose degradation I pathway ($-0.74 \leq R^2 \leq 0.82$; $p \leq 0.01$).

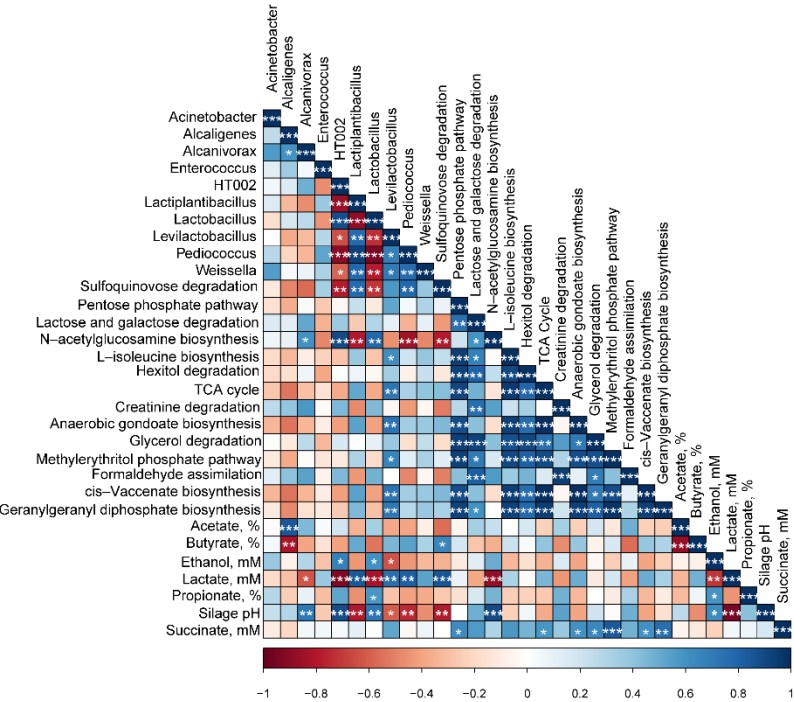

**Figure 3.** Correlation coefficients of bacterial communities and MetaCyc pathways identified by the linear discriminant analysis effect size (LEfSe) tool as significantly affected by vegetable mixture, correlated with common silage fermentation parameters. Positive correlations (blue) and negative correlations (red) are considered significant where * < 0.05; ** < 0.01 and *** < 0.001.

The predicted L-isoleucine biosynthesis III, the citric acid cycle, anaerobic gondoate biosynthesis, methylerythritol phosphate pathway II, cis-vaccenate biosynthesis, and geranylgeranyl diphosphate biosynthesis II (via MEP) pathways were all moderately positively correlated ($R^2 \geq 0.61$; $p \leq 0.03$; Figure 3) with *Levilactobacillus*. However, *Lactobacillaceae* HT002, *Lactiplantibacillus*, *Lactobacillus,* and *Pediococcus* had a strong negative correlation with N-acetylglucosamine biosynthesis pathways ($-0.78 \leq R^2 \leq 0.79$; $p \leq 0.01$). Correlations between bacteria are demonstrated in Figure 3. Briefly, positive correlations were detected between *Alcaligenes* and *Alcanivorax* ($R^2 = 0.59$; $p = 0.04$; Figure 3), while the relative abundance of *Lactobacillaceae* HT002 had a strong negative correlation with *Levilactobacillus* and *Pediococcus* ($-0.90 \leq R^2 \leq -0.65$; $p \leq 0.02$).

### 3.7. Fungal Community Structure and Profile Abundance

There was no effect of level, probiotic, or level × probiotic on the fungal community structure of ensiled samples (PERMANOVA: $R^2$ = 0.12; $p$ = 0.95; Figure 4A). Although, after 14 d aerobic exposure, fungal community structure tended to be similar between 20 and 40% DM vegetable mix but separated from the 0% DM vegetable mix treatment (PERMANOVA: $R^2$ = 0.42; $p$ = 0.08; Figure 4B), which was likely influenced by the number of mini-silos used in this study.

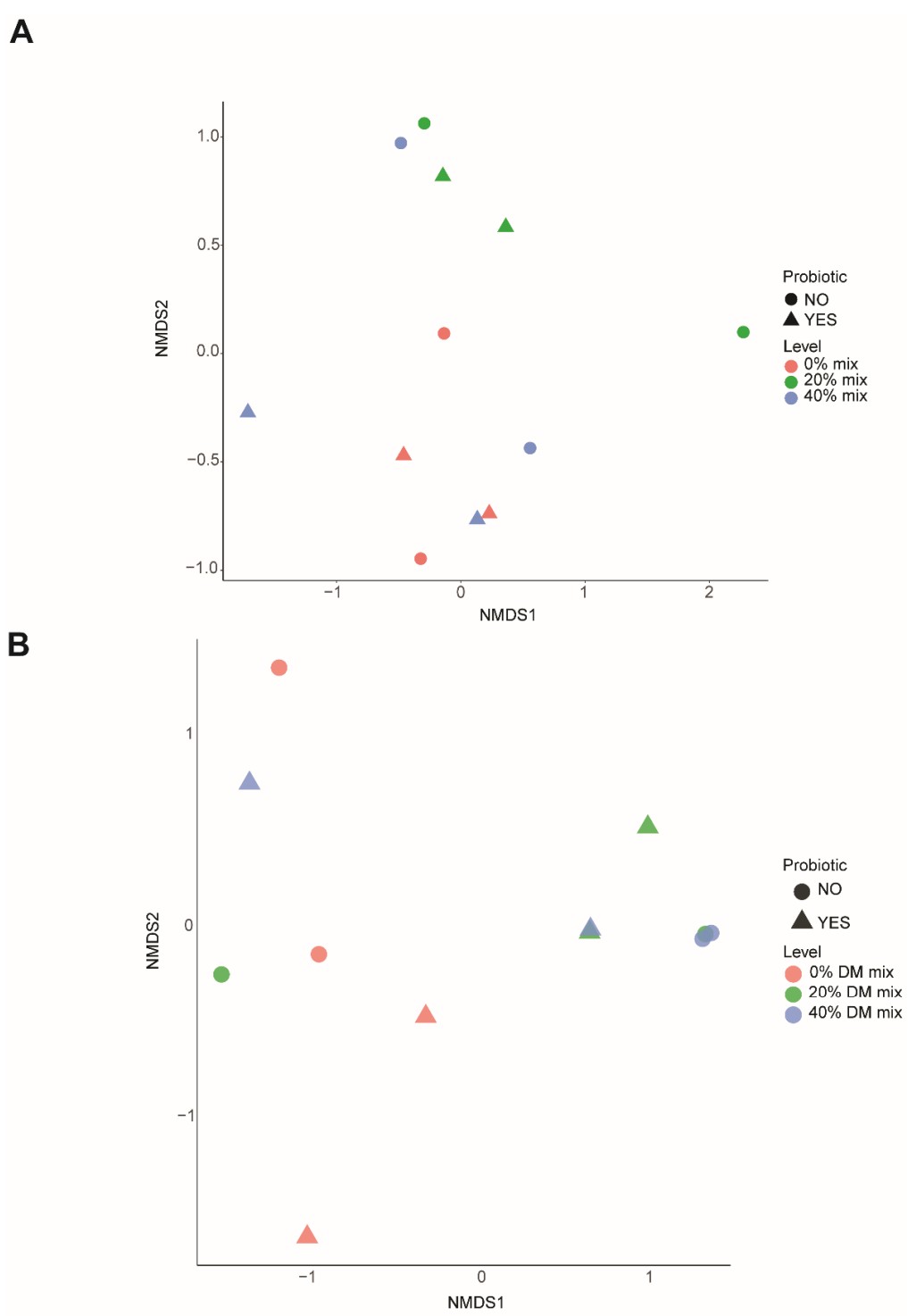

**Figure 4.** Non-metric multidimensional scaling analysis (NMDS) based on the Bray–Curtis dissimilarities for the fungal microbiota of sorghum ensiled with a vegetable mixture at 0, 20 or 40% DM (**A**) after ensiling for 101 d, as well as after (**B**) 14 d aerobic exposure, with or without a probiotic inoculant.

Thirty-one fungal taxa were identified in the microbiota of the silages opened on day 101. Of these taxa, the interaction of level × probiotic reduced ($p$ = 0.01; Table S5) the relative abundance of *Cladosporium exasperatum* from 0.50 ± 0.05% to an undetectable level among uninoculated samples ensiled with 20 or 40% DM mix, and in inoculated samples with 20% DM mix. No other effect of vegetable level, probiotic, or vegetable level × probiotic was observed ($p$ = 0.11) on the relative abundance of other fungal species, except for *Filobasidium magnum*, which was reduced ($p$ = 0.05) from 0.13 ± 0.03% to 0.00% with increasing vegetable mixture. Despite no statistical effect ($p$ ≥ 0.05), *Pseudallescheria ellipsoidea* dominated uninoculated samples ensiled with 20% vegetable mix (97.2 ± 27.2%) and was relatively abundant in the remaining silage samples. *Kazachstania humilis* was also relatively abundant in the microbiota of inoculated 0 and 40% mix silage samples (17.4 and 44.8 ± 19.6% relative abundance).

After 14 d aerobic exposure, seventeen fungal genera, and species were detected within the silage microbiota, with 0% vegetable mix treatments dominated by *Monascus pilosus* (67.7 ± 12.8%; $p$ = 0.05; Table S6), irrespective of probiotic inoculation. No other effect of level, probiotic, or level × probiotic was observed for the silage fungal communities ($p$ ≥ 0.16). Dominant taxa within the silage samples included *K. humilis* (99.8 ± 28.7%; uninoculated 40% DM), *Pichia* spp. (27.2 ± 11.3; uninoculated 20% DM) and *Gibberella circinata* (24.6 ± 7.39%; inoculated 0% DM.

### 3.8. Aerobic Stability

There was no effect ($p$ ≥ 0.10; Table 4) of probiotic or vegetable level × probiotic on silages exposed to aerobic conditions for 14 d. Across d 1 to 14, the inclusion of a vegetable mix at 40% DM reduced silage temperature below room temperature on d 14, compared to 0% DM silage (−0.66 vs. 0.05 ± 0.10°C). No silage exceeded 2 °C above room temperature and was therefore not deemed spoiled at 14 d post opening.

**Table 4.** Temperature differences between the ambient and silage temperature (°C) of sorghum ensiled with 0, 20, and 40% vegetable mixture, with or without a probiotic inoculant (Prob) under exposure to aerobic conditions for 14 d.

| | Level, % DM | | | | *p*-Value | | | | |
|---|---|---|---|---|---|---|---|---|---|
| Day | 0 | 20 | 40 | SEM | Level | Prob | Level × Prob | L | Q |
| 0 | −2.74 | −2.59 | −2.79 | 0.09 | 0.28 | 0.19 | 0.62 | 0.70 | 0.13 |
| 1 | −2.18 [a] | −2.38 [b] | −2.69 [c] | 0.05 | <0.01 | 0.10 | 0.36 | <0.01 | 0.40 |
| 2 | −1.76 [a] | −2.14 [b] | −2.48 [c] | 0.05 | <0.01 | 0.57 | 0.67 | <0.01 | 0.76 |
| 3 | −1.56 [a] | −1.94 [b] | −2.18 [c] | 0.06 | <0.01 | 0.47 | 0.56 | <0.01 | 0.35 |
| 4 | −1.34 [a] | −1.66 [b] | −2.04 [c] | 0.07 | <0.01 | 0.62 | 0.27 | <0.01 | 0.78 |
| 5 | −1.00 [a] | −1.30 [b] | −1.61 [c] | 0.05 | <0.01 | 0.65 | 0.10 | <0.01 | 0.91 |
| 6 | −0.71 [a] | −1.04 [b] | −1.38 [c] | 0.06 | <0.01 | 0.31 | 0.85 | <0.01 | 0.94 |
| 7 | −0.44 [a] | −0.66 [b] | −0.99 [c] | 0.04 | <0.01 | 0.61 | 1.00 | <0.01 | 0.35 |
| 8 | −1.51 [a] | −1.83 [b] | −2.25[c] | 0.06 | <0.01 | 0.70 | 0.07 | <0.01 | 0.42 |
| 9 | −1.33 [a] | −1.46 [a] | −2.00 [b] | 0.06 | <0.01 | 0.91 | 0.12 | <0.01 | 0.04 |
| 10 | −0.55 [a] | −0.61 [a] | −1.41 [b] | 0.12 | <0.01 | 0.80 | 0.48 | <0.01 | 0.02 |
| 11 | −0.34 [a] | −0.50 [a] | −1.24 [b] | 0.10 | <0.01 | 0.68 | 0.32 | <0.01 | 0.06 |
| 12 | −0.20 [a] | −0.29 [a] | −1.13 [b] | 0.10 | <0.01 | 0.55 | 0.32 | <0.01 | 0.02 |
| 13 | −0.01 [a] | −0.14 [a] | −0.84 [b] | 0.11 | <0.01 | 0.95 | 0.50 | <0.01 | 0.09 |
| 14 | 0.05 [a] | −0.01 [a] | −0.66 [b] | 0.10 | <0.01 | 0.56 | 0.79 | <0.01 | 0.02 |

Abbreviations: Only values for Level are presented, as there was no effect of Prob or Level × Prob interaction ($p$ ≥ 0.07); SEM, standard error of the mean. Different letters (a, b, c) within the rows indicate differences ($p$ ≤ 0.05). L, linear; Q, quadratic.

### 3.9. In Vitro Rumen Fermentation, Gas, and Methane Production

A single effect of level × probiotic was observed, where IVDMD increased in uninoculated silages with up to 20% DM vegetable mix (53.3 ± 0.73% vs. 44.7 ± 0.75%; $p$ = 0.04;

Table 5). Similarly, inoculated samples had the greatest IVDMD with 40% DM vegetable mix compared to 0% (50.0 vs. $45.2 \pm 0.75\%$; $p = 0.04$). A tendency ($p = 0.06$) was noted for the effect of level $\times$ probiotic for pH. Further, no effect of probiotic was observed across parameters ($p \geq 0.23$), but ensiling sorghum with 40% DM vegetable mix reduced total gas and $CH_4$ on a mL/g digestible DM basis by 14.5 and 9.6% ($p \leq 0.01$), respectively, compared to the 0% DM vegetable mix silage.

**Table 5.** In vitro fermentation, gas, and methane production parameters of sorghum ensiled with 0, 20, or 40% DM unsalable vegetable mixture, with or without a probiotic inoculant after 24 h in vitro batch culture incubation.

| | Level, % DM | | | | *p*-Value | | |
|---|---|---|---|---|---|---|---|
| | **0** | **20** | **40** | **SEM** | **Level** | **L** | **Q** |
| Total gas, mL | 39.7 | 40.4 | 39.2 | 1.64 | 0.66 | 0.73 | 0.40 |
| Total gas, mL/g DM | 82.4 | 83.8 | 81.6 | 3.38 | 0.68 | 0.75 | 0.42 |
| Total gas, mL/g digestible DM | 182.3 [a] | 157.7 [b] | 155.9 [b] | 6.69 | <0.01 | <0.01 | 0.01 |
| $CH_4$, % | 16.3 | 17.1 | 16.8 | 0.49 | 0.13 | 0.17 | 0.14 |
| $CH_4$, mL/g digestible DM | 29.2 [a] | 27.1 [b] | 26.4 [b] | 0.64 | 0.01 | <0.01 | 0.40 |
| $CH_4$, mL/g incubated DM | 13.1 | 14.0 | 13.4 | 0.33 | 0.16 | 0.53 | 0.07 |
| IVDMD, % | 44.9 [b] | 51.6 [a] | 50.7 [a] | 0.53 | <0.01 | <0.01 | <0.01 |

Abbreviations: Only values for the effect of Level are presented, as there was a single effect of Prob and Level × Prob (IVDMD; $p = 0.04$); IVDMD, in vitro dry matter digestibility; SEM, standard error of the mean. Different letters (a, b) within the rows indicate differences ($p \leq 0.05$). L, linear; Q, quadratic.

### 3.10. In Vitro Rumen Volatile Fatty Acids

In vitro VFA production was largely unaffected by vegetable level × probiotic, except for propionate, which was greater in concentration ($p = 0.01$; Table 6) in uninoculated 40% DM (21.8 vs. $19.9 \pm 0.95\%$) and inoculated 20% DM vegetable mix (21.7 vs. $19.5 \pm 0.95\%$) samples compared to the control. Consequently, a tendency ($p = 0.06$) was noted for the ratio of acetate to propionate, which was largely influenced by vegetable inclusion in 0, 20 and 40% DM inoculated samples (2.92 vs. 2.67 vs. $3.13 \pm 0.12$), respectively. However, total VFA increased by 8.8%, and butyrate reduced by 15.5% with 40% DM vegetable mix ($p \leq 0.01$), while no other effect of the level was observed across in vitro VFA ($p \geq 0.17$).

**Table 6.** Total volatile fatty acids (VFA) concentration and percentages of individual VFA of total VFA in samples of sorghum ensiled with 0, 20, or 40% DM unsalable vegetable mixture, with or without a probiotic inoculant after 24 h in vitro batch culture incubation.

| | Level, % DM | | | | *p*-Value | | |
|---|---|---|---|---|---|---|---|
| | **0** | **20** | **40** | **SEM** | **Level** | **L** | **Q** |
| pH | 6.7 | 6.7 | 6.7 | 0.01 | 0.08 | 0.06 | 0.22 |
| Total VFA, mM | 97.1 [b] | 103.6 [a] | 105.6 [a] | 4.71 | 0.01 | <0.01 | 0.38 |
| *Volatile fatty acids as a percentage of total VFA* | | | | | | | |
| Acetate (A), % | 56.9 | 57.6 | 59.0 | 0.89 | 0.17 | 0.06 | 0.75 |
| Propionate (P), % | 19.7 | 21.2 | 20.5 | 0.86 | 0.04 | 0.15 | 0.03 |
| Butyrate, % | 18.1 [a] | 16.0 [b] | 15.3 [b] | 1.63 | 0.01 | <0.01 | 0.38 |
| BCVFA, % | 3.75 | 3.88 | 3.71 | 0.36 | 0.38 | 0.76 | 0.18 |
| Caproate, % | 0.50 | 0.48 | 0.47 | 0.05 | 0.89 | 0.67 | 0.84 |
| Valerate, % | 1.04 | 0.80 | 0.95 | 0.49 | 0.68 | 0.74 | 0.42 |
| Ratio A:P | 2.89 | 2.72 | 2.88 | 0.10 | 0.17 | 0.57 | 0.08 |

Abbreviations: Only values for the effect of Level are presented, as there was no effect of Prob ($p \geq 0.18$), and a single effect of Level × Prob (propionic acid; $p = 0.01$); VFA, volatile fatty acids; BCVFA, branch chained volatile fatty acids (iso-butyrate + iso-valerate); SEM, standard error of the mean. Different letters (a, b) within the rows indicate differences ($p \leq 0.05$). L, linear; Q, quadratic.

## 4. Discussion

This study evaluated the effect of ensiling sorghum with 0, 20, or 40% DM unsalable vegetable mix on silage chemical composition, fermentation quality parameters, and the associated bacterial, archaeal, and fungal communities. Previous results ([7,34,35]) support the findings from this study that ensiling unsalable vegetables lead to readily degradable silage (increase in IVDMD by 12.9%), dominated by lactic acid bacteria. In addition to ensuring silage quality, the presence of lactic acid bacteria in silages may confer a probiotic benefit to ruminants given their subsistence in the rumen microbial community upon consumption by the ruminant [36,37].

Heterofermentative lactic acid bacteria have the capacity to utilise alternative methods of carbohydrate metabolism to produce lactate, acetate, ethanol, $CO_2$, and 1,2-propanediol during anaerobic fermentation [38,39]. In this study, the relationship identified between HT002, a candidate genus in the *Lactobacillaceae* family, *Lactobacillus*, and *Levilactobacillus* with ethanol, lactate, and silage pH indicates a role for heterofermentative bacteria in the fermentation of sorghum ensiled with 40% DM vegetable mix. Lactic acid bacteria including *Lentilactobacillus buchneri* may catabolise lactate in the production of greater quantities of acetate [40], which is a likely explanation for the decrease of up to 62% lactate with 40% DM vegetable mixture seen after 101 d ensiling in this study. Furthermore, the greatest quantity of lactate was observed after 7 d ensiling, decreasing with the increase in acetate during the remaining ensiling period. The Chao1 estimator of microbial richness in this study increased by 53% after 7 d ensiling with 40% DM vegetable mix, with no impact on diversity. This, along with a corresponding shift in silage pH indicates a transition from more acid-tolerant homofermentative, to less-acid-tolerant heterofermentative bacteria within similar genera, contrasting other reports [41,42]. Although no difference was detected in the concentration of acetate produced after 101 d ensiling, it still comprised the greatest proportion of VFA in this study (approx. 83% total VFA).

The strong negative correlation of the predicted sulfoquinovose degradation pathway with lactic acid bacteria in this study further suggests a shift away from the predominant production of lactate to acetate, as observed in taxa such as *Escherichia coli* [43], *Eubacterium rectale,* and *Bilophila wadsworthia* [44]. Of the vegetable mixture utilised in this study, carrots contain 21.4% sulfoquinovosyl diacylglycerol of total glycolipids [45], a ubiquitous organosulfur compound primarily found in the chloroplast membranes of photosynthetic organisms [46,47]. To our knowledge, the sulfoquinovosyl diacylglycerol content of green beans and onions has yet to be elucidated and was not measured in this study. Regardless, the predicted enrichment of the sulfoquinovose degradation pathway in the silage microbiome at 101 d ensiling suggests an involvement of the bacterial community in the cleavage of sulfoquinovosyl diacylglycerol to sulfoquinovose during ensiling, and consequent catabolism of sulfoquinovose to sulfosugars utilised for bacterial growth [48]. The 12.8% increase in acetate concentration in this study, corresponding with an increase in silage pH from 3.9 ± 0.03 to 4.4 ± 0.03 with 40% DM vegetable mix, has been observed in other studies [49], where an increase in acetate from 16.3 to 55.3 ± 0.26 g kg DM corresponded with a 6% increase in pH in silages treated with heterofermentative inoculants [50].

Ensiling of 40% DM vegetable mix with sorghum was associated with *Lactobacillus* and *Lactobacillaceae* HT002, coinciding with the potential enrichment of metabolic pathways associated with hexitol degradation. It is well documented that hexitol can be utilised as a source of carbon and energy by lactic acid bacteria through the catabolism of lactate [51,52], but these bacteria are dependent on the presence of enzymes such as pyruvate oxidase, pyruvate-formate lyase, or a combination of the two [53]. The prevalence of these pathways characterised in this study may provide further evidence for the function of *Lactobacillaceae* HT002 an uncharacterized candidate genus within the *Lactobacillaceae* family. This is contrary to our hypothesis, where homofermentative *Lactobacillus* was expected to dominate all silage treatments.

Two predicted metabolic pathways (anaerobic gondoate biosynthesis and cis-vaccenate biosynthesis) were positively correlated with *Levilactobacillus* here and were similarly observed in Pennisetum silages [54]. This association is not surprising, considering that gondoate is produced during anaerobic fermentation, and has also been correlated with unclassified *Lactobacillales* ($R^2 = 0.85$; $p < 0.01$; [55]), while both gondoate biosynthesis and cis-vaccenate biosynthesis pathways contribute to unsaturated fatty acid metabolism [56]. The methylerythritol phosphate pathway and resultant geranylgeranyl diphosphate biosynthesis are also traits of Gram-positive bacteria, which includes *Levilactobacillus* [57].

Lactic acid bacteria readily utilise a host of carbohydrates as sources of carbon for growth [58,59], including N-acetyl glucosamine, a monosaccharide of chitin and derivative of glucose [60]. The N-acetyl glucosamine synthesis pathway was positively correlated with taxa of the *Lactobacillaceae* family (*Lactobacillaceae* HT002, *Lactobacillus*) and *Alcanivorax*. Importantly, *Lactiplantibacillus* was the only lactic acid bacteria genus that was negatively correlated with N-acetylglucosamine, indicating potential increased metabolic activity during the ensiling process, given that this genus was dominant in this study. These findings are supported by studies that inoculated triticale silage with *Lactobacillus brevis*, previously demonstrating rapid utilisation of N-acetylglucosamine, indicative of its inhibition of spoilage bacteria and probiotic potential [61]. Inoculation of sugarcane silage with lactic acid bacteria also had similar findings, where 6 of 10 strains isolated rapidly utilised their carbohydrate source [38]. Moreover, 7 of 10 strains of lactic acid bacteria isolated from whole crop wheat silage fully or partially utilised N-acetylglucosamine as a source of carbon and proliferated within a pH range of 4.5–8.0 [62]. The genus *Alcanivorax* comprises three species, one of which utilises N-acetylglucosamine [63] and was detected in microbial communities of silage in this study. Although, *Alcanivorax* is typically detected in marine sediments, and to our knowledge, has not been profiled in prior silage studies.

*Alcanivorax* and *Alcaligenes* spp. were positively correlated and greater in 20% DM mix silages compared to the 0% DM mix silages (undetected vs. $0.16 \pm 0.07\%$ and $0.13 \pm 0.05\%$ respectively) in our study. Members of these genera conduct the first stage of heterotrophic nitrification of ammonia into nitrite and have been detected in bacterial communities in the initial stages of nitrification [64]. During nitrification, the degradation of urea and subsequent release of ammonia by *Alcaligenes* in maize silage increases silage pH [65], which provides an environment conducive to fungal growth. The presence of *Alcaligenes* in maize silages has previously coincided with the dominance of *Aspergillus* [66], but this was not predominant in this study. Given that *Aspergillus* was not detected in any silage fungal communities, we speculate that it may have been inhibited by the production of antifungal volatiles by *Alcaligenes* taxa [67] given the resistance of vegetable mix silages to aerobic spoilage. However, further investigation is required to fully elucidate the antifungal properties of *Alcaligenes* taxa.

Co-fermentation of lactate with the degradation of glycerol, a pathway predicted to be enriched in 20% DM vegetable mix silages has been noted in species such as *L. buchneri* and *L. brevis* [68], reducing the net production of lactate by conversion of pyruvate to acetate. Integral to the citric acid cycle, pyruvate may also be utilised as a precursor to citrate, an intermediate produced by the donation of an acetyl group from acetyl-CoA to oxaloacetate [69]. The presence of *Lactobacillaceae* taxa during the ensiling process likely contributes to the aerobic stability of resultant silages, as inoculation of sorghum with *L. buchneri* decreased lactate concentrations while simultaneously increasing the concentration of acetate and silage pH [70,71]. As observed in this study, the numerically greater quantity of acetate (77.6 vs. $87.5 \pm 3.47\%$) could likely have contributed to the lower temperature difference, thus greater aerobic stability after 14 d aerobic exposure in 40% DM vegetable mix silages.

A numeric increase in acetate concentration with increasing vegetable mix also corresponded with greater silage pH, a trait similarly observed during incubation days 7 to 30 in isolated *L. plantarum* at a pH of 5.0 [72]. Comparatively, acetate and ethanol produced by heterofermentative lactic acid bacteria, or by *Alcaligenes* spp. as an end-product of

metabolism [73,74] increases silage aerobic stability as noted by increasing vegetable inclusion level up to 40% DM mix in this study. After 14 d aerobic exposure, the atmospheric temperature did not exceed 21 °C, despite *Monascus* spp. proliferating at temperatures between 25 to 37 °C and a pH of 2.5–8 [75]. To our knowledge, few reports have detected *Monascus* spp. below 25 °C, while a paucity of information exists regarding the characteristics of *Monascus pilosus* in silage fungal communities. Comprising one of the three most relatively abundant taxa, the greatest relative abundance of *Monascus* spp., particularly *M. pilosus* was in 0% DM silages, irrespective of probiotic inoculation, at the end of the ensiling period, and after 14 d aerobic exposure. Although, the silage environment may have provided opportune conditions for growth, thus dominance of *M. pilosus* was considered a decrease of up to 67% for both richness (Chao1) and diversity (Shannon) measures with increasing vegetable inclusion.

Moreover, given the greater proportion of soluble carbohydrates in the vegetable mixture silages, the dominance of *M. pilosus* was expected, as the predominant mode of growth of *Monascus* spp. is via lytic enzymes, utilising mono- and disaccharides, starch, and pectin as proliferation substrates [76]. However, spoilage at the silage surface likely occurred, as aerobic exposure increased the relative abundance of *M. pilosus* up to $44.4 \pm 18.1\%$ in 40% DM vegetable mix silages.

After ensiling and the aerobic exposure period, *K. humilis* was a dominant taxon in fungal communities of 40% DM vegetable mixture silages, irrespective of probiotic, a result similarly observed when maize ensiled with 40% DM pumpkin was subjected to aerobic exposure [6]. Metabolising glucose as a growth substrate, *K. humilis* added to sourdough fermentation with *Saccharomyces cerevisiae* has previously increased the production of lactate, acetate, and ethanol [77]. *P. ellipsoidea* is typically detected in soil, and its presence could indicate soil contamination during sorghum harvest, but little is known about its function [78]. Further, this study did not predict the function of fungal communities in vegetable mix silages, given that significant sequence features are not shared between fungi that diverge and specialise to their host [79].

## 5. Conclusions

Ensiling sorghum with an unsalable vegetable mixture, irrespective of inoculation with a probiotic produced a silage that was dominated by heterofermentative taxa within the family *Lactobacillaceae*. This presence was evident through the differential abundance of metabolic pathways in the predicted metagenomes associated with alternate carbohydrate metabolism and characterised by reduced lactate, with no impact on acetate. Therefore, pathways of alternative carbohydrate metabolism facilitated by heterofermentative lactic acid bacteria can increase the feed value of sorghum when ensiled with an unsalable vegetable mixture at 40% DM, without requiring a high quantity of lactate.

**Supplementary Materials:** The following supporting information can be downloaded at: https://www.mdpi.com/article/10.3390/fermentation8120699/s1, Table S1: Values for pH, lactate, succinate, ethanol, acetate, and total VFA concentrations from sorghum ensiled with an unsalable vegetable mixture at 0, 20 or 40% DM, with or without a probiotic inoculant. Silages were sampled during the ensiling period after 0, 1, 3, 5, 7, and 101 days (D) ensiling. Data for D101 silages influenced by level ($p \leq 0.05$) are displayed in the manuscript (Table 2). Table S2: Alpha diversity metrics of archaeal and bacterial communities from sorghum ensiled with an unsalable vegetable mix at 0, 20, or 40% DM, with or without a probiotic inoculant. Silage samples were collected at 0, 1, 3, 5, 7, and 101 days (D) ensiling, including 14 d aerobic exposure. Silages sampled at D0, D101, and D115 that were influenced by Level ($p \leq 0.05$) are displayed in the manuscript (Table 3). Table S3. Alpha diversity metrics of fungal communities from sorghum ensiled with an unsalable vegetable mix at 0, 20, or 40% DM, with or without a probiotic inoculant. Samples were ensiled over 0, 1, 3, 5, 7, and 101 days ensiling, including 14 d aerobic exposure. Silages sampled at D0, D101, and D115 that were influenced by Level ($p \leq 0.05$) are displayed in the manuscript (Table 3). Table S4. Bacterial and archaeal genera in the microbiota of sorghum ensiled with 0, 20, or 40% DM unsalable vegetable mix with or without a probiotic inoculant. Values expressed as a % relative abundance. Table S5. Fungal

genera and species in the microbiota of sorghum ensiled with 0, 20, or 40% DM unsalable vegetable mix with or without a probiotic inoculant. Values expressed as a % relative abundance. Table S6. Fungal genera and species in the microbiota of sorghum ensiled with 0, 20, or 40% DM unsalable vegetable mix with or without a probiotic inoculant after exposure to aerobic conditions for 14 days. Values expressed as a % relative abundance.

**Author Contributions:** Conceptualisation: D.L.F., S.J.M. and A.V.C.; Methodology: D.L.F., S.J.M. and A.V.C.; Software: D.L.F. and A.V.C.; Validation: D.L.F., A.V.C., D.B.H.; Formal Analysis: D.L.F., S.J.M., A.V.C. and D.B.H.; Investigation: D.L.F., S.J.M. and A.V.C.; Writing Original Draft Preparation: D.L.F.; Writing Review & Editing: D.L.F., S.J.M., A.V.C. and D.B.H.; Visualisation: D.L.F.; Supervision: S.J.M. and A.V.C. All authors have read and agreed to the published version of the manuscript.

**Funding:** This research did not receive any specific grant from funding agencies in the public, commercial or not-for-profit sectors.

**Institutional Review Board Statement:** The study was conducted according to the Animal Care and Protection Act (2001) and the Animal Care and Protection Regulation (2012) and approved by The University of Queensland Animal Ethics Committee (Approved Protocol Number 2021/AE000823 and Monday, 27 September 2021).

**Informed Consent Statement:** Not applicable.

**Data Availability Statement:** The 16S rRNA and ITS1 gene sequences are available through the NCBI sequence read archive under BioProject accession PRJNA821258.

**Acknowledgments:** The authors would like to thank Meat & Livestock Australia for their financial support of Daniel Forwood. Additionally, we would like to thank Kalfresh Pty Ltd. for supplying the carrots, green beans, and onions used in this study. Finally, the authors thank Génome Québec; for the sequencing and details of the methodology.

**Conflicts of Interest:** The authors declare no conflict of interest.

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
