# Peer review of "Ensiled Mixed Vegetables Enriched Carbohydrate Metabolism in Heterofermentative Lactic Acid Bacteria"

_fermentation, doi:10.3390/fermentation8120699_

Round 1

Reviewer 1 Report

1.    The exact sources and information of LAB inoculum should be detailed in the manuscript.

2.    Line 345 : Please apply italic style to microbial species through the manuscript

3.    Regarding to the originality of the manuscript in the last paragraph (Introduction), it should be indicated clearly how this manuscript contributes to the existing knowledge. Please also identify and describe the originality of the study described in this manuscript and how it will likely to contribute to the state-of-the-art

4.    The contents of the relationship with lactic acid bacteria should be further reinforced in the introduction part.

5.    The materials used and the details and conditions of experimental procedures have to be described with sufficient clarity

6.    Line 74: Please check the superscript

7.    Figure 1 : prob/probiotic ; please check your manuscript to make sure all abbreviations are used correctly

Author Response

Responses to the comments:

  1. We disagreed with the reviewer. Abstract is written following the authors guideline.
  2. Thank you. This was corrected for clarity. However, the limitations of the Abstract word count were a consideration of Abstract composition.
  3. Please see comment above, there were limitations on the Abstract.
  4. Thank you, this has been addressed and was an error during typesetting.
  5. Thank you. However, the presentation of references are in line with the Fermentation Instruction to Authors.
  6. Thank you for your comment. Primers were listed in the manuscript as requested (LN133-135).
  7. Thank you for your comment. We request more information on this information. What is meant by organic composition analysis?
  8. Thank you. This was corrected and reflected in the manuscript.
  9. Thank you for the observation. This was corrected in Table 1 and consequently in Table 2.
  10. Thank you. Image size was intended to be larger in publication and were reduced in size during typesetting.

Author Response

Responses to the comments:

  1. Done, thank you.
  2. Done, thank you.
  3. Thank you. This is required, as this demonstrates the role of two separate lactic acid bacteria, and would be incorrect otherwise.
  4. Done, thank you.
  5. Done, thank you.
  6. LN76 was edited for clarity
  7. LN74 was edited for clarity.
  8. Thank you, sections were combined for flow.
  9. It is assumed that ensiling time will influence samples as fermentation by lactic acid is well-documented. The study focused on mixed vegetable and probiotic, so samples were collected on each day to reflect this.
  10. Thank you for your comment. Respectfully, where the probiotic had no effect this was not included for brevity and to limit the number of tables in the manuscript. Often the effect of the vegetable mix, which was the focus of the study was more prominent, thus these effects were included in the manuscript.
  11. Thank you for your suggestion. Correlation analysis of this nature is not appropriate for this study, given the silage should be fed prior to spoiling. Aerobic stability was recorded to assess the silage shelf-life after exposure to aerobic conditions.
  12. We disagreed with the reviewer. Discussion does discuss the biological means of the main results. Thank you.

Reviewer 3 Report

This is good work with well discussed results. The Ms is interesting, however, it contains many errors and unclear parts that need to be corrected.

o   The abstract is confusing, and did not show the clear cut results of this Ms in term of comparison between the treatments and the final conclusion as gad been shown in conclusion section

o   L19, correct to the 0% DM   and please show that vegetable mix is 0%  to remove confusion

o   From L19-L27, should be rephrased and show the results and impacted data

o   -Almost all formation of microbial names are incorrect all should be italic in the text and reference section (too many to list starting from L52 to the end of Ms), L77-80

o   The presentation of references should be displayed first by author names before the numbers in some sentence such as  for example (the method of [7], L67, L93, L113, 117, 122, 129 and others

o   -L29 show the sequences of the used primers

-rephrase the sentence (The DaDA2v1.8[17]....etc  to be clear for the readers

o   The authors should provide the organic composition analysis of the vegetable mix used in this study with the % ratio of each compound

o   Formatting of CH4 and kg/m3 should be corrected

o   Table 1 the authors did not show the pH values as indicated in the title; Column 1 please write vegetable mix level in all tables to be clear

o   Correct the formatting of scientific names at fig 2, and fig 3 (please enlarge the figures to be more clear)

Author Response

(The authors gave the same response as above.)

Round 2

Reviewer 1 Report

The answer to the previous question (No. 1) is not clear.

Please record bacterial culture collection number. For example, ATCC 15036,,,,

1. The exact sources and information of LAB inoculum should be detailed in the manuscript 

Author Response

Revision 2

Reviewer 1

Responses to the comments:

The answer to the previous question (No. 1) is not clear.

Response: We did address the Question 1 from reviewer 1 on revision 1. The product information for the Bio Soil probiotic used in this study n provided in the manuscript (lines 77-82)

Please record bacterial culture collection number. For example, ATCC 15036,,,,

Response: This information cannot be provided. We did not culture any bacteria. As described in lines 77-82, the probiotic used was commercial and we provided all the information from their website as well as on the product label. The company no longer exist.

The exact sources and information of LAB inoculum should be detailed in the manuscript 

Response: As described in lines 77-82 probiotic used was commercial and we provided all the information from their website as well as on the product label. The company no longer exist.

Manuscript has been checked by 3 native English speakers and there are no spelling errors (checked by proof reading as well as Microsoft Word Spelling check).

Reviewer 2 Report

The author need to relate the results obtained during this study with relevant discussion and compare the results obtained to previously results from other researchers with more specifically the effect of vegetables mix and probiotics inoculant.

Author Response

Revision 2

Reviewer 2

Responses to the comments:

The author need to relate the results obtained during this study with relevant discussion and compare the results obtained to previously results from other researchers with more specifically the effect of vegetables mix and probiotics inoculant.

Response: This is a very novel study. As far as the authors are aware of, there are no other literature on mix veggies silages with probiotics like ours. The purpose of the discussion section is to interpret and describe the significance of your findings in relation to what was already known about the research problem being investigated and to explain any new understanding or insights that emerged as a result of our research. We wrote our discussion in that way. Our discussion is connected to the introduction by way of the research questions we posed. Our discussion does not simply repeat or rearrange the first parts of our paper; our discussion clearly explains how our study advanced the reader's understanding of the research problem from where we left them at the end. Our discussion also discusses the biological meaning of our results.